# Targeting the Granulocytic Defense against *A. fumigatus* in Healthy Volunteers and Septic Patients

**DOI:** 10.3390/ijms24129911

**Published:** 2023-06-08

**Authors:** Stefanie Michel, Lisa Kirchhoff, Peter-Michael Rath, Jansje Schwab, Karsten Schmidt, Thorsten Brenner, Simon Dubler

**Affiliations:** 1Department of Anesthesiology and Intensive Care Medicine, University Hospital Essen, University Duisburg-Essen, Hufelandstraße 55, D-45147 Essen, Germany; stefanie.michel@uk-essen.de (S.M.); jansje.schwab@uk-essen.de (J.S.); karsten.schmidt@uk-essen.de (K.S.); thorsten.brenner@uk-essen.de (T.B.); 2Institute of Medical Microbiology, University Hospital Essen, Hufelandstraße 55, D-45147 Essen, Germany; lisa.kirchhoff@uk-essen.de (L.K.); peter.rath@uk-essen.de (P.-M.R.); 3Institute of Medical Microbiology, University Hospital Essen, Excellence Center for Medical Mycology (ECMM), Hufelandstraße 55, D-45147 Essen, Germany

**Keywords:** invasive pulmonary aspergillosis, *Aspergillus fumigatus*, neutrophil granulocyte, critically ill, cortisol, host-pathogen interactions, sepsis

## Abstract

Neutrophil granulocytes (NGs) are among the key players in the defense against *Aspergillus fumigatus* (*A. fumigatus*). To better elucidate a pathophysiological understanding of their role and functions, we applied a human cell model using NGs from healthy participants and septic patients to evaluate their inhibitory effects on the growth of *A. fumigatus* ex vivo. Conidia of *A. fumigatus* (ATCC^®^ 204305) were co-incubated with NGs from healthy volunteers or septic patients for 16 h. *A. fumigatus* growth was measured by XTT assays with a plate reader. The inhibitory effect of NGs on 18 healthy volunteers revealed great heterogeneity. Additionally, growth inhibition was significantly stronger in the afternoon than the morning, due to potentially different cortisol levels. It is particularly interesting that the inhibitory effect of NGs was reduced in patients with sepsis compared to healthy controls. In addition, the magnitude of the NG-driven defense against *A. fumigatus* was highly variable among healthy volunteers. Moreover, daytime and corresponding cortisol levels also seem to have a strong influence. Most interestingly, preliminary experiments with NGs from septic patients point to a strongly diminished granulocytic defense against *Aspergillus* spp.

## 1. Introduction

*Aspergillus fumigatus* (*A. fumigatus*) is a ubiquitous mold. Even though people may inhale up to 100,000 *Aspergillus* conidia (spores) daily, it is not a critical issue for healthy and immune-competent individuals [1]. Clearance of inhaled *Aspergillus* conidia is the result of a well-coordinated immune response involving epithelial cells, alveolar macrophages, and neutrophil granulocytes (NGs) [2].

Patients who are immunosuppressed (e.g., after solid organ transplantation) [3], and especially those with neutropenia (e.g., after allogeneic stem cell transplantation) [4], are at high risk of developing IPA (invasive pulmonary aspergillosis) [5]. In the last decade, new groups of immunocompetent patients without classical risk factors were discovered to be at risk for IPA: patients with viral disease (especially influenza or coronavirus) and patients with sepsis or septic shock [6]. Unfortunately, the IPA-associated mortality rate remains as high as 80% [7].

Neutrophil granulocytes (NGs) are among the most important immune cells in the defense against *A. fumigatus*. NG depletion after an *A. fumigatus* challenge resulted in fatal infections in immunocompetent mice [8]. The chemotaxis of NGs through CXC chemokines is a critical function to control infections caused by *Aspergillus* spp. Preclinical models for IPA have shown that deficiency of CXCR (receptor)-2 or CXCL1/2 impairs neutrophil chemotaxis. Vice versa, severe respiratory infections can alter chemokine expression, leading to decreased resistance against *A. fumigatus* [9,10,11]. NGs are crucial for *A. fumigatus* elimination. The production of reactive oxygen species (ROS) and the formation of neutrophil extracellular traps (NETosis) are primary effector mechanisms [10]. Genetic defects in nicotinamide adenine dinucleotide phosphate oxidase (NADPH oxidase complex) lead to impaired ROS production, resulting in an inability to kill microorganisms [12].

Sepsis is defined as “life-threatening organ dysfunction caused by a dysregulated host response to infection” [11]. Even though sepsis leads to a fast and massive proinflammatory response, patients develop a state of immunosuppression in parallel or shortly thereafter [9]. Regarding the relevant mechanisms, NGs isolated from septic patients show multiple functional modifications, including impaired chemotaxis (decreased expression of CXCR1 and CXCR2) and migration properties [9,10,13]. Moreover, the defense mechanism (oxidative burst) is impaired, hampering the clearance of pathogenic fungi or bacteria [13]. Additionally, an increased presence of immature NGs during sepsis leads to significant T-cell suppression, which is associated with early mortality after sepsis [14].

IFNs and NGs play key roles in antifungal defense. Additional studies are urgently needed to elucidate the complicated interplay between viral and fungal infections, on the one hand, and sepsis and fungal infections, on the other. A better understanding of the immune mechanism of the patient’s host defense against *A. fumigatus* during sepsis or viral infection could pave the way to identifying potential new pathways. These could be targeted by immune-based therapies to work alongside new antifungal drugs to decrease the high mortality rate among these patients.

Despite the various advantages of murine models (easy to manipulate, rapid reproductive rate, and low cost), they will never be able to perfectly reflect influenza +/− IPA or sepsis in humans [15,16]. In the work presented here, we characterize the NG-derived defense against *A. fumigatus* in a human cell model using NGs from healthy volunteers and septic patients. Our approach provides the basis for a deeper understanding of the interplay between NGs and *A. fumigatus* in critically ill patients with sepsis or viral infections.

## 2. Results

### 2.1. Concentrations of NGs and Aspergillus Conidia and Their Ratio Ex Vivo

The starting points were co-incubation of 1 × 10^5^ and 1 × 10^6^ NGs (effector, E) with 1 × 10^6^
*A. fumigatus* conidia (target, T), corresponding to E/T ratios of 10:1 and 1:1, respectively (Appendix A). At E/T ratios of 1:1, 10:1, and 20:1, NGs did not show any inhibitory effect after 3 and 6 h. Further increasing the E/T ratio to 640:1 and 1.280:1 revealed an inhibitory effect of NGs compared to *A. fumigatus* only (Appendix A). Quantification of the inhibitory effect was only possible visually under the microscope.

### 2.2. Duration of Co-Culture with NGs and A. fumigatus Conidia

As assessed by microscopy, swollen conidia and germ tubes were found; however, there was almost no hyphal growth in co-culture of NGs and *A. fumigatus* conidia (E/T ratio 10:1) after 3 and 6 h (Appendix A). In an overnight assay (23 h), dense mycelium formed (Appendix A). The co-culture time was then shortened to 16 h, and the following results were observed: The hyphae started growing but did not build dense mycelium. At this point in the study, interpreting the inhibitory effect of NGs was only possible visually through the microscope, whereas quantitative analysis was not yet possible.

In the next experimental step, an XTT assay was used for quantification of *Aspergillus* growth inhibition using 96-well cell culture plates (Sarstedt AG and Co. KG, Nümbrecht, Germany) and DMEM D5030 (Sigma Aldrich, St. Louis, MO, USA) in an incubator with CO_2_. An increase in the E/T ratio led to decreased growth of *A. fumigatus*. E/T ratios of 10:1 and 20:1 showed almost no inhibitory effect (2% and 4% growth inhibition), while 2560:1 and 5125:1, on the other hand, inhibited *A. fumigatus* growth by 46% and 61%, respectively. See Figure 1a for different E/T ratios of NGs versus Aspergillus conidia after 16 h and *A. fumigatus* only. Here, 156 and 312 Aspergillus conidia in combination with 2 × 10^5^ NGs per well turned out to be optimal (E/T ratios of 1280:1 and 640:1, respectively) for XTT reproducibility.

### 2.3. Heterogeneity of Aspergillus Growth Inhibition by NGs from Healthy Participants

The function of NGs from 18 healthy volunteers was assessed after 16 h of co-culture at an E/T cell ratio of 1280:1 (equaling 156 conidia) by cell lysis, FACS, and XTT measurements. In the sample without cell lysis (Figure 1b), NGs could be detected (blue spots), but there were no NGs left after lysis (Figure 1c). Even after 16 h of co-culture in vitro, a small proportion of NGs were still alive and could be stimulated by zymosan (Figure 1d,e). The mean age of the volunteers was 32 years (±6.4 years (SD)). The NG count range in whole-blood samples was 1.5 × 10^6^ to 7.7 × 10^6^/mL (mean 3.3 × 10^6^/mL).

Heterogeneous results were found for *Aspergillus* growth inhibition by NGs from different healthy volunteers (Figure 2a). The mean inhibition rate was 38.7% (minimum 8.3%, maximum 94.2%). Figure 2b shows the mean growth inhibition of *A. fumigatus* (E + T) versus *A. fumigatus* only (control).

### 2.4. NG Functions at Different Times of the Day and Corresponding Cortisol Levels

To narrow down the reasons for the observed heterogeneity of *A. fumigatus* defense among healthy volunteers, we repeated the experiments at different times of the day (7 a.m. and 4 p.m.). NGs from eleven healthy volunteers (five men, six women; mean age 32 years; mean leucocyte count 7.21 × 10^6^/mL; mean NG count 3.2 × 10^6^/mL) were analyzed. Growth inhibition was significantly stronger in the afternoon (4 p.m., mean 57%) than in the morning (7 a.m., mean 53%) (mean difference 13.8%, *p* = 0.0028; Figure 2c). The corresponding cortisol levels were physiologically higher in the morning than in the afternoon (mean at 7 a.m., 605 nmol/L vs. 4 p.m., 296 nmol/L).

### 2.5. Aspergillus Growth Inhibition by NGs from Patients with Sepsis/Septic Shock

NGs from patients with sepsis/septic shock (n = 5) according to the latest definitions [17] were challenged with *A. fumigatus*. NGs from five healthy subjects served as controls. NGs from patients with sepsis/septic shock revealed decreased growth inhibition (median 34%) compared to NGs from healthy controls (median 45%) (Figure 2d). The five patients were 43.8 years of age (mean) with a body mass index (BMI) of 27.78 m^2^. All patients were male and suffered from severe acute respiratory distress syndrome (ARDS) according to the latest Berlin definition [18]. Healthy controls were 35.4 years of age (mean), with a BMI of 24.4 m^2^. Three controls were male, and two were female.

## 3. Discussion

Critically ill patients with sepsis or septic shock and patients with viral infections are prone to invasive fungal infections, which are associated with high morbidity and mortality rates. Studies highlight the importance of NGs in the context of antifungal defense, but the complicated interplay between NGs and fungi in this patient group remains elusive. Most studies in this field use animal models, so the results cannot be transferred to humans without limitations [8,19,20,21,22,23].

Many studies performed on human cells show impaired neutrophil activity against mold after sepsis [14,24] or viral infections like SARS-CoV-2 [25,26] or influenza [27]. Fungal diseases during viral infections are still associated with a worse outcome. In order to establish faster immune-based diagnostics and treatment strategies, we demonstrate evidence of the granulocytic defense against *A. fumigatus* in healthy volunteers and septic patients using an ex vivo approach. Our first set of experiments used co-incubation of NGs and *Aspergillus* conidia for 3–6 h but failed to show an inhibitory effect due to severely unrestricted hyphae growth in general. Comparable results were reported by Gazendam et al. [28], which might have been due to the fact that resting *Aspergillus* conidia cannot be detected by human NGs. Resting conidia are known to have thick cell walls with limited expression of pattern recognition receptors (PRRs) [29] to prevent detection by NGs. Therefore, we incubated *Aspergillus* conidia alone for 3 h until the appearance of so-called “swollen” conidia. Swollen conidia express a distinctive PRR pattern on their surface [19,29], so they can be more easily detected by human immune cells, such as macrophages or NGs. The co-incubation time was then extended up to 23 h, at which point the inhibitory effects could not be quantified anymore due to mycelium formation.

In our experience, 16 h seemed to be the best duration for co-incubation since the inhibitory effect was found to be maximal after this period. In murine models, the half-life of NGs has been estimated to be only around 12 h. Under homeostatic conditions in vivo, on the other hand, the lifespan of NGs can exceed 5 days [30].

The E:T ratio is of particular importance for the methodological quality and transferability of the data. Our experimental setup used high E:T ratios. This might relate to physiological encounters in vivo, since during the early phase of *A. fumigatus* infection, high numbers of neutrophils are recruited, mediate the early innate immune response, and are considered a key cell population in host defense. As early as 6 h following exposure to *A. fumigatus*, the number of infiltrating NGs increased four times [8,20]. The initial response of NGs to *A. fumigatus*, which orchestrates further immune activation, greatly determines the cause of infection. Therefore, this step can be considered particularly important regarding clinical severity and convalescence. Due to the initial peak of infiltrating NGs, the E:T ratio (NGs: *A. fumigatus*) can be expected to be shifted towards NGs in the early phase of the infection, indicating the need for a high E:T ratio to adequately reflect the clinical situation.

The results for ratios of NG (effector, E) to swollen *Aspergillus* conidia (target, T) of 1:1, 2:1, and 20:1 did not differ from those of the controls (*A. fumigatus* only). NGs in these proportions were obviously not able to prevent swollen conidia from germinating hyphae or damaging existing hyphae. Only E/T ratios above 40:1 showed measurable inhibitory effects. The literature reports a wide range of E/T ratios in co-culture experiments. Morton et al. [31] used dendritic cells and *Aspergillus* hyphae at an E/T ratio of 1:1 in their model, whereas Loeffler et al. [32] worked with germinating *Aspergillus* conidia and human monocytes at an E/T ratio of 2:1. Gazendam et al. [28] used an E/T ratio up to 2.000:1 (*Aspergillus* conidia and NGs) in an overnight assay. Importantly, the endpoints and experimental settings in those studies were fundamentally different from those in our experiments, which means they are hardly comparable. Additionally, different immune cells interact differently and elicit variable immune responses [19].

In a study by Zarember et al. from 2007 [33], polymorphonuclear (PMN) leukocytes from 10 healthy donors were co-incubated with *Aspergillus* conidia (at E/T ratios of 0.5:1 to 3:1) for 16–18 h. Interestingly, even after *Aspergillus* conidia were exposed to PMN for only 2 h, greater amounts of PMN vs. conidia resulted in less conidial and hyphae growth, and this effect was more pronounced in inhibiting hyphae than conidia. However, the authors did not specify whether there was a relevant interindividual difference between donors, as observed in the study.

To the best of our knowledge, this study is the first to show great heterogeneity among NGs from healthy volunteers, all of whom had a good sense of well-being. Several factors are known to influence NG function, including one night of acute sleep deprivation, which is followed by the production of more immature NGs [34]. Immature NGs are known to express decreased levels of the CXCR2 receptor, which is responsible for NG chemotaxis and ROS production. Said et al. [35] showed that chronic sleep deprivation led to a significant decrease in phagocytic activity and decreased the ratios of CXCL10/CXCL9 and CCL5/CXCL9. On the other hand, physical exercise was linked to stimulation of phagocytic activity after even a single training session in another study [36] and higher levels of catecholamine and lower levels of cortisol in a further study [37]. Since all of our healthy participants were medical students or residents, who tend to experience sleep deprivation and lack physical exercise, this might have influenced their NG performance. However, the healthy volunteers did not provide a detailed anamnesis of their sleep and training conditions prior to study inclusion.

The aforementioned results might be linked to changes in hormones (catecholamine and cortisol) due to stress. Cortisol shows anti-inflammatory and immunosuppressive effects in general [38]. Circadian rhythm-induced glucocorticoids are associated with suppressed production of CXCL5 (an NG chemoattractant) and NG recruitment in lung inflammation [38]. Glucocorticoids are important for the development and proliferation of NGs in the bone marrow as well as the inhibition of spontaneous apoptosis in human NGs.

In our study cohort of healthy volunteers, cortisol levels were physiologically higher in the morning than in the afternoon. Accordingly, the ability to inhibit the growth of *Aspergillus* conidia and hyphae was decreased in all volunteers in the morning. Even though corticosteroids have pleiotropic effects on immune cells in different tissues, glucocorticoids can suppress some important functions (e.g., ROS production and chemotaxis) in NGs that are crucial for effectively inhibiting *Aspergillus* conidia and hyphae. To further prove that there is a real causal link between high cortisol levels and reduced fungal growth inhibition by neutrophils, we want to add cortisol to the afternoon co-cultures in the next experimental design to abolish the differences in dependency on time of day (and other potential confounders, including biochemistry parameters).

The first experiments with NGs from septic patients (n = 5) revealed impaired granulocytic defense against *A. fumigatus* compared to healthy volunteers. Although these experiments need to be re-evaluated in a larger cohort of patients with sepsis/septic shock, these findings are in line with the recent literature, which clearly describes impaired NG functioning in sepsis/septic shock. In a study by Cox et al., septic patients were characterized by decreased NET formation [39]. In a study by Schenz et al. [40], NG energy metabolism was drastically impaired during sepsis, potentially leading to a hampered granulocytic defense against infection. Additionally, NGs lacking functional IFN-λ receptors showed substantially impaired ROS production, resulting in insufficient antifungal activity [41]. Which of the functional limitations of septic NGs in the investigation presented here (phagocytosis, chemotaxis, ROS, or NET formation alone) might have led to the reduced granulocytic defense against *Aspergillus* spp. needs to be re-evaluated in future experiments.

Our study has several limitations. As an ex vivo experimental setup, the transferability of the results to an in vivo setting may be limited. Important determinants of the immune reaction against *A. fumigatus*, including further immune cells, cytokines, and tissue-specific characteristics, were not recapitulated in the experimental setup. The E:T ratio may have inadequately reflected the pathophysiological situation, leading to an overestimation of the observed effect. We did not differentiate between the effects of the killing of *Aspergillus* conidia and the containment of *Aspergillus* hyphae since NGs use different mechanisms to accomplish these activities [28]. However, even after 16 h of co-culture in vitro, a small proportion of NGs were still alive and could be stimulated by zymosan. These NGs might have contributed to the inhibition of *Aspergillus* hyphal growth. Although we always used the same stock solution of *A. fumigatus*, the fungus could have functioned differently on different occasions.

## 4. Materials and Methods

### 4.1. Inclusion Criteria

Healthy volunteers ≥18 years of age were included in this study after approval was received from the Institutional Review Board of the medical faculty of the University of Duisburg-Essen (20-9700-BO, 20-9352-BO). All volunteers were employees of the department of anesthesia at the University Hospital Essen and were non-smokers with a body mass index (BMI) of <30 kg/m^2^. No volunteers were on daily medication. Healthy was defined as having no history of previous illnesses. Adult patients with sepsis or septic shock were also included in the study. Sepsis was defined according to the latest definition (Sepsis-3) [17].

NG counts were measured during recruitment, and none of the participating volunteers were found to have neutropenia (defined as an NG count of <1.5 × 10^6^/mL). The study was conducted in the Department of Anesthesiology and Intensive Care Medicine and the Institute of Medical Microbiology, University Hospital Essen, University of Duisburg-Essen (Essen, Germany), between December 2020 and August 2022.

### 4.2. Media and Reagents

Dulbecco’s Modified Eagle Medium (DMEM), pH 7.4 (Sigma Aldrich, St. Louis, MO, USA), with NaCl, N-2-hydroxyethylpiperazine-N-2-ethane-sulfonic acid (HEPES) 5 mM (Thermo Fisher, Waltham, MA, USA), pyruvate 1 mM (Thermo Fisher), D-glucose 10 mM (Sigma Aldrich, Darmstadt, Germany), L-glutamine 2 mM (Sigma Aldrich), and 10% FCS in superior heat-inactivated form (Biochrom S0615; Sigma Aldrich)Triphenyltetrazoliumchloride (XTT) (Thermo Fisher, Waltham, MA, USA) at a final concentration of 0.2 mg/mL with 50 µM menadione (Sigma Aldrich). XTT is a colorless salt that is reduced to an orange metabolite by vital cells.

### 4.3. Cultivation of A. fumigatus

*A. fumigatus* (ATCC^®^ 204305; Thermo Fisher Scientific, Waltham, MA, USA) was cultivated at 37 °C on malt agar (MEA) for 2 days, then harvested with 0.1% Tween 20 (Sigma Aldrich, St. Louis, MO, USA) in deionized water. The supernatant was filtered (Filcon, BD Biosciences, Franklin Lakes, NJ, USA), washed twice with cold Dulbecco’s phosphate-buffered saline (DPBS) (Thermo Fisher), and stored at 4 °C in DMEM. The concentration of the stock solution was calculated microscopically (40× magnification; Zeiss Axiocam, Carl Zeiss AG, Oberkochen, Germany) with a Neubauer counting chamber and by counting colony forming units (CFU) (40× magnification; Zeiss Axiocam). CFU plates on MEA were used to control the concentration. In our final standardization protocol before co-culture with NGs, *Aspergillus* conidia were pre-incubated at 37 °C for 3 h for maturation and swelling to stimulate the NG response through changes in surface proteins [42].

### 4.4. Isolation of NG

The human NGs were isolated with a MACSxpress^®^ Whole Blood Neutrophil Isolation Kit (Miltenyi Biotec, Bergisch Gladbach, Germany) according to the manufacturer’s protocol. The remaining erythrocytes were removed using the Ery Depletion Kit^®^ (Miltenyi Biotec). Purified NGs were counted with an XP-300™ cell counter (Sysmex Cooperation, Kōbe, Japan). NGs were directly used for co-culture after cell count.

### 4.5. Quantitative Measurement of Cell Vitality

The XTT assay (Cell Signaling Technology, Danvers, MA, USA) is a color metric method used to measure cell vitality. Vital cells reduce colorless tetrazolium to orange formazan. The viability is equivalent to the intensity of the orange metabolite and can be measured by a plate reader [43]. The assay was performed according to the manufacturer’s protocol. Briefly, XTT and menadione were added to the cells, which were incubated without light for 2 h [43]. After two cycles of centrifugation at 2000 RCF, the absorption of the supernatant was measured at a wavelength of 492 nm through a plate reader. The integrated optical density (IOD), which is the reduced transmission intensity of light when the sample is being illuminated [44], was assessed. More fungal growth leads to a denser sample. *Aspergillus* growth inhibition was then calculated as follows: [(IOD of *A. fumigatus* only) − (IOD of *A. fumigatus* + NGs)/(IOD of *A. fumigatus* only)] × 100. All controls and test samples were pipetted as eight repetitions in a 96-well chamber. The mean of these eight repetitions was used for calculation. Normalization was performed with the light absorption of a control containing the medium and *Aspergillus* conidia only undergoing the exact same procedure. Therefore, 0% growth inhibition was defined as the same light absorption in the control sample without NGs and in the experimental sample containing NGs. Cell lysis of NGs was performed to eliminate absorption by NGs. The study protocol of the co-incubation experiments and the timeline of the final standardized protocol using XTT are presented in Figure 3.

### 4.6. Lysis of Cells

After 16 h of co-incubation, the cultured NGs were lysed via three washing steps with deionized H_2_O, followed by 40 min of incubation in deionized H_2_O (400 rpm, RT) and centrifugation (2000 RCF, 5 min). Flow cytometry staining buffer was applied (Thermo Fisher, Waltham, MA, USA), and fluorescence-activated cell sorting (FACS) analysis of the NG control was used to confirm NG lysis.

### 4.7. Microscopic Analysis

The microscopic analysis was initially performed with a Zeiss Axiocam microscope (Carl Zeiss AG, Oberkochen, Germany) at 10×, 20×, and 40× magnification. Co-culture was performed for 3, 6, 18, and 24 h with RPMI medium and fetal bovine serum in chamber slides (Thermo Fisher, Waltham, MA, USA). Calcofluor white (BD, Franklin Lakes, NJ, USA) and acridine orange staining (Sigma Aldrich, St. Louis, MO, USA) were applied according to the manufacturers’ protocols. The inhibitory effect was assessed visually. Finally, native microscopy (Zeiss Axio microscope, 20× magnification, no staining) of samples from each co-culture was performed in parallel with XTT quantification to check the quality of the experiment visually. E:T ratios and duration of co-culture were optimized in the preceding experiments.

### 4.8. FACS Analysis

NGs were stimulated with zymosan (37 °C for 30 min) and stained with CD11b (PE) antibodies. Sytox Green was applied for confirmation of NG vitality. FACS analysis (Beckmann Coulter Life Sciences, Krefeld, Germany) [45] with sideward and forward scattering was subsequently performed to identify NG cells by size and granularity.

### 4.9. Statistical Analysis

The data were analyzed using GraphPad Prism (GraphPad, San Diego, CA, USA). Continuous data were tested for normal distribution using the Shapiro-Wilk test. Normally distributed data were analyzed using the Student’s *t*-test. The Mann-Whitney U-test was employed for non-normally distributed data. Data that did not meet this criterion were analyzed using ANOVA. All analyses had exploratory intentions, and *p* < 0.05 was considered significant.

## 5. Conclusions

The cell model showed a great heterogeneity of NGs from healthy volunteers regarding the defense against *A. fumigatus.* Moreover, this granulocytic defense seemed to be relatively diminished in NGs from patients with sepsis or septic shock. Future experiments will have to re-evaluate to what extent the immune defense mechanism of NGs against *A. fumigatus* is hampered (e.g., via NET formation, ROS production, the NADPH oxidase system, or chemotaxis) in larger cohorts of patients with sepsis, septic shock, or viral infections. This could pave the way toward developing new immune-based therapies and help reduce the high IPA mortality rate.

## Figures and Tables

**Figure 1 ijms-24-09911-f001:**
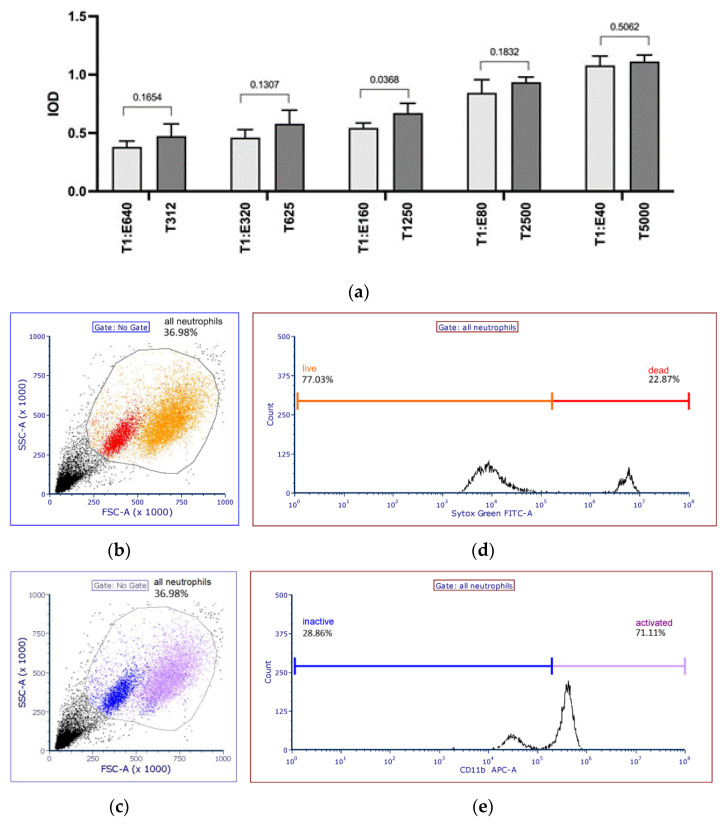
(**a**) Dilution of co-cultures and controls measured using XTT and a plate reader. (**a**) Co-culture of NGs from one healthy participant at different E/T ratios after 16 h of co-incubation with corresponding controls with A. fumigatus only in different concentrations. (**b**–**e**) Gating strategy for neutrophil viability and activity. Prior to FACS analysis, purification of the neutrophils was controlled with an XP-300™ cell counter (Sysmex Cooperation, Kōbe, Japan). To identify the dead cell population, a mix of living and dead cells is shown here. For the dead cell controls, cells were killed at 60 °C for 10 min. (**b**,**c**) Scatter plot profiles show back gating to confirm gating strategy. (**d**,**e**) Histograms to evaluate the intensity of the Sytox Green staining or the relative expression of CD11b activation. Only neutrophils gated in scatter plots were used for the histograms. Cells were backgated onto FSC vs. SSC to identify specific neutrophil populations. FITC: fluorescein isothiocyanate; NG: neutrophil granulocyte.

**Figure 2 ijms-24-09911-f002:**
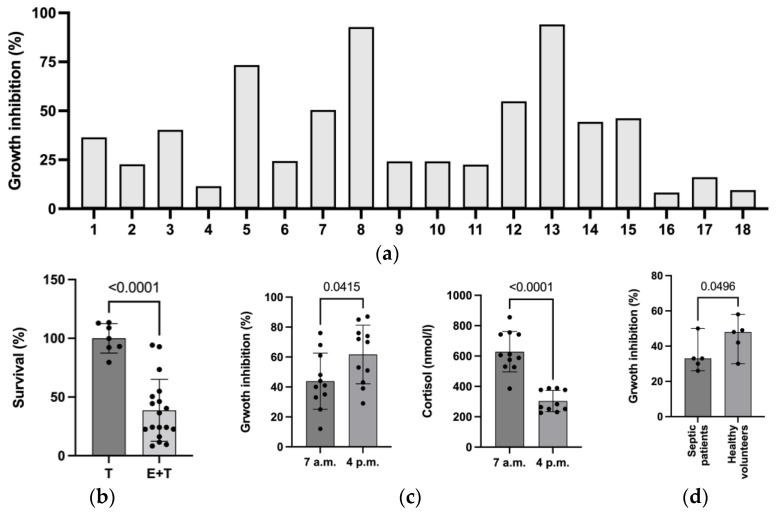
(**a**) NG co-culture of 18 healthy volunteers measured using XTT and a plate reader. NGs and *A. fumigatus* (E/T = 1280:1) after 16 h of co-incubation. (**b**) Summary of co-culture (E + T) results of all 18 healthy volunteers with an E/T ratio of 1280:1 after 16 h of co-incubation. “Survival” was defined as growth inhibition. Results are presented as means. T represents the controls. (**c**) NG co-culture of 11 healthy participants measured using XTT and a plate reader at different times of the day (7 a.m. and 4 p.m.). Cortisol serum levels (nmol/L) were measured simultaneously. (**d**) Summary of co-culture (E + T) results of 5 septic patients compared to 5 healthy subjects with an E/T ratio of 1280:1 after 16 h of co-incubation. Results are presented as medians. Abbreviations: IOD = integrated optical density; NG = neutrophil granulocytes; T = target; E = effector.

**Figure 3 ijms-24-09911-f003:**
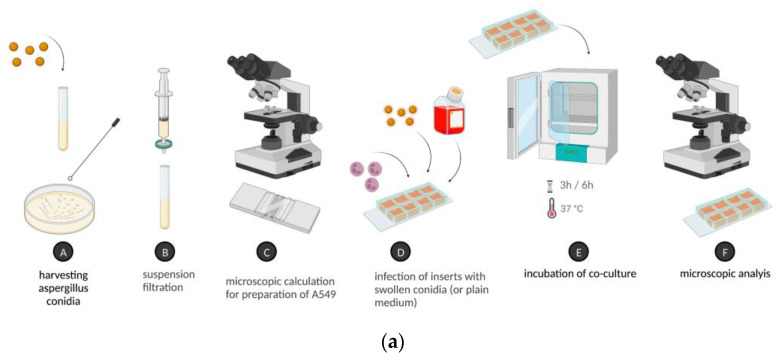
(**a**) The study protocol of co-incubation experiments. (**b**) Timeline of the final standardized protocol using XTT for quantitative analysis. Figures illustrated with BioRender^©^. NG, neutrophil granulocyte.

## Data Availability

Due to privacy or ethical restrictions, the data presented in this study are available on request from the corresponding author.

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
