# Peer review of "Targeting the Granulocytic Defense against A. fumigatus in Healthy Volunteers and Septic Patients"

_ijms, 2023, doi:10.3390/ijms24129911_

Round 1

Reviewer 2 Report

Michel et at. study interactions of neutrophils from healthy donors and few sepsis patients with Aspergillus fumigatus. Although studies of impaired anti-mold immunity in sequential infection scenarios (e.g., post-sepsis, post-viral) are of major interest, this manuscript adds little to the literature, given the flawed data presentation (e.g., overstating the novelty of the methodology), highly artificial assay conditions (e.g., extremely high effector/target ratios), very low numbers of data sets for the comparison of sepsis patients and healthy controls, and superficial/unidimensional readout portfolio. Therefore, this manuscript fails to make a meaningful contribution to our understanding of post-sepsis aspergillosis. Currently, the only interesting finding in this paper is the circadian effect on neutrophilic killing of A. fumigatus and even that is not entirely novel and limited by the artificiality of the methodology. 

A major concern is that the manuscript is presented as a “assay/model development” paper. However, the novelty is overstated, as various XTT assays protocols for studies of neutrophil/Aspergillus interactions have been published and used in the past. It seems, that the differences in the authors’ protocol are rather minor, not warranting such a strong focus on the methodological aspects of their study (including in the title).

Moreover, the authors’ assays are performed at extreme E:T ratios. I am concerned that these highly artificial assay conditions are not representative of real-life in-vivo conditions. Did the authors check the viability and functionality of their neutrophils with other methods (e.g., ability to elicit oxidative burst)? Neutrophils are hugely susceptible to pre-analytic delays, minor handling issues, or rapid changes in temperature (e.g., cold wash buffers). Therefore, there could be many procedural confounders explaining poor antifungal activity of the cells, necessitating huge E:T ratios. 

Figure 1 has no scientific value and should be relegated to the supplement or deleted. It would suffice to state in Materials & Methods that E:T ratios and duration of co-culture were optimized in preceding experiments.

Panels 2a is currently pointless, as it needs to display the co-culture conditions and corresponding “Aspergillus only” controls side by side, along with significance testing.

Panels 2b-2e should be shown either in Materials & Methods or in combination with the corresponding quantitative results.

Panel 3a should be presented as a box/whisker plot with individual data points, as done in panels 3b-d.

Panel 3d is pointless with n = 2 vs. n = 3. Why did the authors only show 2 healthy controls? How were they selected? In panels 3a-3c, they have analyzed 18 healthy controls. At the very least, they could have compared the 3 patients and 18 healthy controls.

Lines 257-268: The authors clearly acknowledge that others were able to see substantial fungal killing at much more adequate (lower) E:T ratios. However, their premise that these studies differed fundamentally from the methodology used in the present paper is flawed. There are many studies that used very similar XTT assays to study interactions of neutrophils and Aspergillus, including in patients with prior viral or bacterial infections (e.g., https://pubmed.ncbi.nlm.nih.gov/36052094/), and found considerable fungal killing (and significant differences between patients and controls) at low E:T ratios. This clearly suggests that the assays presented in the current paper are methodologically flawed and not properly performed and/or optimized.

Minor issues:

Lines 46-54: Given that the authors did not show chemokine secretion data in their manuscript, the introduction of chemokines is too detailed and unnecessary.

Lines 237-238: This statement disregards the many studies performed on human ex-vivo cells that have shown impaired neutrophil activity against molds after sepsis or viral infections, based on larger sample size and proper methodology, including the reference provided above.

The authors should resize the figures to avoid page breaks within figures. It would also be advisable to put the figures and corresponding captions on the same page.

The manuscript needs language editing by a native speaker. Some sentences (including in the abstract) are wordy and the manuscript contains several weird German-English words/expressions (e.g., “concentration row”, “a good sense of wellbeing”).

Round 2

Reviewer 1 Report

Satisfied with the revisions made by the authors. 

Some editing of the English language is rquired. 

Author Response

Dear Editor, Dear Reviewer

Thank you very much for assessing our manuscript. We are very happy that our manuscript is now potentially acceptable for publication in the “International Journal of Molecular Science” after implementation of the latest Reviewers` suggestions.

If you have any further questions or suggestions, please do not hesitate to contact us.

Thank you very much for your helpful suggestions and for your kind support.

With kind regards

Simon Dubler

Reviewer 2 Report

The authors have performed substantial revisions and provided a detailed response in their rebuttal letter. The quality and conciseness of the writing have been improved substantially and some of the scientific concerns (e.g., low numbers of subjects/patients) have been partially addressed. Although I still have some concerns regarding high E:T ratios and the limited significance and novelty of the findings, the manuscript has now become a "publishable" unit that others might find useful. However, I have 2 comments that should be addressed:

1.) The authors have included an extensive justification for the use of high E:T ratios in their rebuttal letter. However, they should incorporate at least 2-3 sentences in the actual manuscript (Discussion) describing why they believe there is value in using high E:T ratios and how that might relate to physiological in vivo encounters.

2.) Throughout the manuscript, including the abstract (line 21-22), the association between high cortisol levels and reduced fungal growth inhibition by neutrophils is presented as a proven causal link. Although an obvious assumption, this is, however, not proven by the authors' data (e.g., a multivariate analysis taking into account several other confounders and biochemistry parameters). Therefore, these statements should be attenuated.

English language editing clearly improved the readability of the manuscript. 
